# Osteonecrosis of the Femoral Head: A Multidisciplinary Approach in Diagnostic Accuracy

**DOI:** 10.3390/diagnostics12071731

**Published:** 2022-07-16

**Authors:** Adrián Cardín-Pereda, Daniel García-Sánchez, Nuria Terán-Villagrá, Ana Alfonso-Fernández, Michel Fakkas, Carlos Garcés-Zarzalejo, Flor María Pérez-Campo

**Affiliations:** 1Departamento de Bioquímica y Biología Molecular, Facultad de Medicina, Universidad de Cantabria-IDIVAL, 39012 Santander, Spain; daniel.garcias@alumnos.unican.es; 2Servicio de Anatomía Patológica, Hospital UM Valdecilla, Universidad de Cantabria, 39008 Santander, Spain; nuriateran@unican.es; 3Servicio de Traumatología y Ortopedia, Hospital Universitario Marqués de Valdecilla-IDIVAL, Universidad de Cantabria, 39008 Santander, Spain; anaalfonso@scsalud.es (A.A.-F.); michelfakkas@scsalud.es (M.F.); carlosgarces@scsalud.es (C.G.-Z.)

**Keywords:** avascular necrosis of the hip, diagnosis, radiological tests, reproducibility, reliability, accuracy, histopathology

## Abstract

Osteonecrosis of the Femoral Head (ONFH) is a disabling disease affecting up to 30,000 people yearly in the United States alone. Diagnosis and staging of this pathology are both technically and logistically challenging, usually relying on imaging studies. Even anatomopathological studies, considered the gold standard for identifying ONFH, are not exempt from problems. In addition, the diagnosis is often made by different healthcare specialists, including orthopedic surgeons and radiologists, using different imaging modes, macroscopic features, and stages. Therefore, it is not infrequent to find disagreements between different specialists. The aim of this paper is to clarify the association and accuracy of ONFH diagnosis between healthcare professionals. To this end, femoral head specimens from patients with a diagnosis of ONFH were collected from patients undergoing hip replacement surgery. These samples were later histologically analyzed to establish an ONFH diagnosis. We found that clinico-radiological diagnosis of ONFH evidences a high degree of histological confirmation, thus showing an acceptable diagnostic accuracy. However, when the diagnoses of radiologists and orthopedic surgeons are compared with each other, there is only a moderate agreement. Our results underscore the need to develop an effective diagnosis based on a multidisciplinary approach to enhance currently limited accuracy and reliability.

## 1. Introduction

Osteonecrosis of the Femoral Head (ONFH) is a disabling condition that usually results in the collapse of the femoral head and secondary osteoarthritis (OA) in young adults and middle-aged individuals, with a mean age of presentation of 38 years [1]. To date, there is no epidemiological report on ONFH worldwide, however, some countries have already performed studies on the incidence of this disease. In the United States the number of new cases each year is estimated to be greater than 30,000 [2], with these numbers steadily increasing yearly. Importantly, ONFH seems to be the direct cause of between 5 and 18% of all the hip arthroplasties performed annually in the United States [3]. Apart from the important healthcare costs associated with the surgical treatment of the ONFH, the fact that this disease causes severe pain and disability mainly in adults at a productive age also translates into an important socioeconomic burden [4]. Clinical presentation of ONFH is generally asymptomatic in early stages, although occasionally patients could indicate hip or groin pain. At initial stages, negative plain radiographs are common, therefore, ONFH must be suspected if patients present any of the reported risk factors [5]. Thus, a precise diagnosis and staging of ONFH to treat our patients optimally becomes critical. Presumably, the optimal therapeutic approach might include a multimodal treatment regime, or a patient tailored plan from hip surgery to rehabilitation [6,7,8].

In the early stages the disease can be treated with conservative approaches, such as drugs, biological therapies, or extracorporeal shock wave therapy, to delay or stop its progression. Unfortunately in many cases this condition will lead to loss of integrity of the subchondral bone, requiring surgical treatments, typically a total hip arthroplasty within two years after the development of hip pain [9]. To avoid this outcome, an early and accurate diagnosis that would allow us to treat this pathology in its early stages is key [10]. However, the early diagnosis of ONFH is highly challenging, as the onset of symptoms and imaging characteristics are insidious and subtle [11], frequently posing a diagnostic problem to orthopedic surgeons. The difficulty associated with ONFH diagnosis often results in the presentation of advanced cases of this disease, when femoral-head-conserving surgical treatment is no longer indicated [12].

Longitudinal studies of patients with various forms of osteonecrosis and osteochondritis show that the history of mild cases of ONFH (abnormal alterations in soft tissues observable only by MRI) is to naturally heal, with only the most severe cases requiring a hip arthroplasty and being susceptible to histological studies [13]. Ideally, therefore, ONFH diagnosis should be made by non-invasive means, such as imaging techniques. However, to be able to perform an accurate diagnosis using these non-invasive tests, an adequate correlation with the gold standard, the anatomopathological exam [13], should be clearly stablished. ONFH diagnosis currently relies largely on imaging techniques. The imaging study is usually initiated by a common radiographical analysis. However, as previously mentioned, plain radiographs are usually negative in early disease, since only a minor osteopenia might be present at this stage. If the findings are inconclusive, more imaging techniques, such as magnetic resonance imaging (MRI), computerized tomography (CT), or bone scan are normally requested. MRI is considered the method of choice for ONFH diagnosis, with the highest sensitivity and specificity [1]. In recent decades, various scientific societies have developed different classification systems for grading the evolution of the process [14]. However, despite these efforts, the controversy regarding the optimal way for classifying osteonecrosis of the femoral head remains. Apart from data such as the etiology, age, occupation, or hip functionality, clearly clarifying the staging of the osteonecrosis is also key in the diagnosis as well as in developing treatment strategies.

The fact that the diagnosis of the ONFH can be performed by different specialists, using different diagnostic tools adds to the challenge of the diagnosis itself. The aim of this paper is to clarify the association and accuracy of ONFH diagnosis between all the professionals involved in its diagnosis including orthopedic surgeons, radiologists, and pathologists.

## 2. Materials and Methods

### 2.1. Study Design

This was a prospective case-control study conducted at the Marqués de Valdecilla University Hospital, in which femoral heads of image-diagnosed ONFH and OA patients undergoing arthroplasty were collected from the operating room for 3 years, to later perform a confirmatory histological analysis. The study sample was determined following the guidelines of other studies in this line made in the field of osteonecrosis [15,16,17,18,19,20,21].

### 2.2. Patients

Patients who met the following criteria were included: (1) between 40 and 90 years old, (2) clinico-radiological diagnosis of ONFH or hip OA, (3) indication of hip replacement surgery [2,22]. Exclusion criteria included: (1) systemic glucocorticoids and/or bisphosphonates treatment history, (2) past or present heavy alcohol consumption, (3) hip trauma or radiation history, (4) storage disorders, pancreatitis, hemoglobinopathies, or dysbarisms history; as we intended to study idiopathic cases of AVNH, with no obvious etiologic factor [10,23] (secondary osteonecrosis).

All patients gave informed written consent. Study protocol was approved by the institutional review board (Comité de Ética en Investigación Clínica de Cantabria, February 2018). Identification Code 2018.014.

### 2.3. Diagnosis

Our study population was triply diagnosed by experienced and certified orthopedic surgeons, radiologists, and pathologists. Orthopedic surgeons diagnosed patients according to signs and symptoms, imaging studies, and macroscopic evaluation of femoral head specimens from the operating room. Plain radiographs and/or MR images were reported by general and musculoskeletal radiologists. Radiographic scoring by the Ficat and Arlet system was used.

From all femoral head specimens macroscopically affected, cylindrical samples underwent anatomopathological analysis. From the histological point of view, a positive diagnosis of ONFH was considered as the presence of significant and diffuse trabecular necrosis (>50% of empty osteocytic lacunae) and necrotic hematopoietic bone marrow in the absence of other specific lesions [18]. For a better histological characterization, other degenerative signs, such as fatty infiltration and medullary fibrosis or heterotopic ossification presence, were graded in qualitative manner (“1-absence”, “2-presence”, “3-moderate”, and “4-intense”) [16].

### 2.4. Histological Analysis

Samples of bone were isolated from the macroscopically affected areas of femoral head specimens (Figure 1A), sectioned in the coronal plane. Bone cylinders were extracted from the sample with the help of a trephine. These samples were preserved and subsequently used for the anatomopathological analyses. For preservation, the cylindrical biopsies were immediately placed in 10% neutral buffered formalin at room temperature for 24 h and subsequently decalcified in 10% ethylenediaminetetraacetic acid at room temperature following standard procedures [24,25]. Decalcified specimens were embedded in paraffin and paraffin sections (4 mm thick) were cut using conventional methods. Sections were stained with Mayer hematoxylin and eosin and Masson trichrome [15,16]. The total number of empty osteocyte lacunae was quantified by two unaware examiners in four non-overlapping regions of interest. At least 300 osteocyte lacunae were quantified in each specimen.

### 2.5. Statistical Analysis

Statistical comparison was performed using chi-square and Fisher’s exact test analysis for categorical data, Student’s t-test for mean comparison (for normally distributed variables), Spearman’s coefficient for rank correlation, the Mann–Whitney U test to compare two related samples (for non-normally distributed variables), and Cohen’s kappa coefficient for inter-rater and intra-rater reliability. A P value probability threshold of less than 0.05 was considered statistically significant. All statistical analyses were performed using SPSS software (SPSS Inc, Chicago, IL, USA).

## 3. Results

Between 2017 and 2020, a total of sixty femoral head specimens were collected from patients undergoing hip replacement surgery at the Marqués de Valdecilla University Hospital. Of those sixty samples, twenty-six corresponded to patients that have had a previous diagnosis of ONFH and thirty-four of them, used as controls in this study, corresponded to patients that have been diagnosed with OA. Clinico-radiological diagnosis was confirmed by anatomopathological analysis which was considered the definitive reference diagnosis. Of all the collected samples, only twelve ONFH and twelve OA specimens met further selection criteria, that is, to have a radiologist-reported imaging study, and a correctly preserved bone sample with enough quality to perform an appropriate histological analysis. Samples lacking a certified radiologist report or showing poor histological quality were excluded.

Histological examination of the femoral head specimens collected confirmed radiological diagnosis in twenty of the twenty-four hips analyzed (84%) (Table 1). Regarding the samples with a diagnosis of ONFH that comply with our selection criteria, only one of the twelve radiologically diagnosed cases of ONFH (8%) was not confirmed by anatomopathological analysis. Interestingly, we found that three of the twelve samples with a radiological diagnosis of OA, representing a 25% of the OA analyzed samples, met the histological diagnosis criteria for ONFH. Furthermore, only 25% of the patients misdiagnosed had been subjected to an MRI scan, highlighting the early reported superior diagnostic capability of this technique. This result, however, could not be considered statistically significant.

A statistically significant difference in the number of empty osteocytic lacunae (t = 5.13; *p* < 0.05) was found between the ONFH (mean ± SE = 68.16 ± 10.57) and the control group (mean ± SE = 40.91 ± 15.03) (Figure 1B and Table 2). There was also a statistically significant difference (U = 35; *p* < 0.05) between medullary fibrosis (Figure 1C) presence between the ONFH (mean ± SE = 2.75 ± 1.21) and the control (mean ± SE = 1.66 ± 0.98) group (Table 2). The histological analysis of other degenerative signs, such as fatty infiltration and heterotopic ossification, indicated no statistically significant differences between the two groups analyzed.

Importantly, the association between the number of empty osteocytic lacunae with the Ficat and Arlet staging could not be considered statistically significant (Table 3). When the diagnoses of radiologists and orthopedic surgeons were compared with each other, twenty-one of the twenty-four diagnoses agreed (k = 0.58; percentage agreement of 85.71%), expressing moderate agreement. Interobserver diagnosis reliability did not differ significantly between the different Ficat and Arlet stages.

For orthopedic surgeons, the rate of agreement between the pre-surgery and the post-surgery diagnosis was 91.67% (kappa value for intra-observer reproducibility of 0.75), expressing substantial agreement.

When the diagnoses of radiologists and orthopedic surgeons were compared with each other, twenty-one of the twenty-four diagnoses agreed (k = 0.58; percentage agreement of 85.71%), expressing moderate agreement. Interobserver diagnosis reliability did not differ significantly between the different Ficat and Arlet stages.

## 4. Discussion

Early diagnosis of ONFH is key to achieving satisfactory therapeutic results that allow prompt selection of an effective joint-preserving treatment. However, accurate diagnosis of ONFH is challenging, especially at the early stages. Normal plain radiographs and physical exams can be falsely reassuring and delay appropriate referral [11,26]. In this context, several studies have found suboptimal diagnostic accuracy, reproducibility, and reliability [13,14,25,27,28,29]. The aim of this paper is to specifically clarify the association and accuracy of ONFH diagnosis between orthopedic surgeons, radiologists, and pathologists, considering imaging modes, microscopic and macroscopic bone features, and disease staging.

Non-specific initial imaging findings and confusing prevalence data are probably the most important causes of ONFH diagnosis disregard. In addition, imaging pitfalls and lack of anatomopathological analysis consensus led to misdiagnosis. Regarding histological diagnosis, there are no established quantitative criteria (number of blocks, trabeculae, or lacunae to quantify) and surgeons’ requisition forms usually lack important information. Therefore, there is a need for a consensus definition of the histological features of ONFH and improvement of samples collection [13].

Empty osteocytic lacunae evaluation is a legitimate method for ONFH diagnosis, not always related directly to age and without a clear relationship with disease stage. In our study, there was a statistically significant difference in the number of empty osteocytic lacunae found between the ONFH and the control group, thus contributing to validate our samples and their histological diagnosis. However, in the present work, the association between age and empty osteocytic lacunae, widely described in the literature [15,30], could not be considered statistically significant. These findings might be explained by the fact that the distribution of empty lacunae in relation to age is quite heterogeneous, compared to the complete loss of osteocytes observed in ONFH [30]. On the other hand, no statistically significant difference was found between the number of empty osteocytic lacunae and Ficat and Arlet stage for ONFH [20]. Although several studies have shown the link between imaging features and osteonecrosis severity [17,20,31], a direct relationship between the proportion of empty osteocytic lacunae and radiological staging has not yet been reported.

The presence of dense medullary fibrosis is significantly greater in necrotic femoral heads when compared to osteoarthritic ones. These results agree with previous findings that describe this degenerative sign in advanced stages of the disease [16,17,20]. In addition, necrosis of the fatty marrow was consistently present in most of the ONFH samples. This result is part of the histological features spectrum of osteonecrosis, with fatty and haemopoietic marrow becoming ghosted [16,30]. Other non-specific degenerative signs examined, such as fatty infiltration and heterotopic ossification, showed no statistically significant difference between both groups (Appendix A).

Importantly, three of the twelve osteoarthritis controls (25%) met the histological diagnosis criteria for ONFH. This finding agrees with previous works, in which 21–31% of cases of ONFH were seen pathologically but not radiographically [27,32]. A study investigating the presence of secondary osteonecrosis in osteoarthritis of the hip confirmed it microscopically in 38.2% of the femoral heads, identifying two different histological patterns: ‘shallow’ osteonecrosis (probably pressure necrosis as a result of eburnation) and ‘deep wedge-shaped’ osteonecrosis (a less frequent, independent phenomena related to primary osteonecrosis) [33].

The ONFH diagnosis varies among medical specialties, and it is not uncommon to find literature failing to mention which methods were used for the diagnosis. Previous works have evidenced a distribution of ONFH diagnosis by specialty of 13% for pathologists (pathology report), 15% for orthopedic surgeons (clinical record), and 19% for radiologists (radiology report) [13]. In our specimen selection, we found a proportion in ONFH diagnosis of 26% for pathologists, 46% for surgeons, and 26% for radiologists.

ONFH diagnosis relies mostly on the assessments performed by the orthopedic surgeon, involving physical examination, medical history, and plain radiographs, or on the radiologist report [27]. However, histological, or even gross pathologic evaluation of hip arthroplasty specimens, is not consistently practiced in medical centers, since the implementation of those analyses is not considered cost-effective [4]. Thus, clinical diagnosis of ONFH remains most of the time unconfirmed. In the hips, a concordance rate of 81.2% in clinical diagnosis verified histologically has been reported [4]. According to the literature, in ONFH this concordance rate varies from 68% to 93% [4,27,31,32]. In the present study, histologic examination of the femoral head specimens confirmed clinico-radiological diagnosis of ONFH in 84% of the cases, hence showing correlation with previous reports. On the other hand, we found that 25% of samples with a radiological diagnosis of OA met histologic diagnosis criteria for ONFH. Previous works have evidenced this kind of false positive in 16% of cases. Misdiagnosis of ONFH can occur if clinician is unaware of potential pitfalls, such as persistent hematopoietic red marrow, the fovea centralis or synovial herniation pits, or existence of pathologic processes that can mimic osteonecrosis, such as subchondral cysts, transient osteoporosis, insufficiency subchondral fractures, osteochondral lesions, and metastasis [10]. Additionally, in agreement with previous reports, we verified that MRI improves diagnostic performance and reduces misdiagnosis of ONFH, with the highest sensitivity and specificity compared to plain radiographs, computed tomography, or scintigraphy. MRI is also highly effective in depicting the early stage and staging lesions accurately [9].

In the present study, when diagnoses of radiologists and orthopedic surgeons were compared with each other, there was only a moderate agreement. Twenty-one of the twenty-four diagnoses agreed (interobserver kappa reliability coefficient of 0.58; percentage agreement of 85.71%), expressing moderate agreement. Literature regarding ONFH interobserver reliability has focused mainly on imaging staging correlation, with average kappa values ranging from 0.39 to 0.56, thus evidencing a poor interobserver reliability, especially among the intermediate stages [25,29,34,35]. In the present work, interobserver diagnosis reproducibility did not show a statistically significant difference when considering the different Ficat and Arlet stages.

Reported evaluations of intra-observer diagnosis variation in ONFH, based mainly in disease imaging staging, have evidenced fair reproducibility, with mean kappa values ranging from 0.43 to 0.88 [25,29,34,35]. Regarding histology, we are not aware of any studies that have looked at intra- or inter-observer variability in the pathological diagnosis of avascular necrosis of the femoral head. This diagnosis was based on macroscopic features of the resected femoral head, an information usually overlooked in literature. Preservation of femoral head sphericity, the presence of degenerative changes or identification of necrotic areas are some of the useful data that a gross evaluation in the operating room can provide, thus contributing to a better diagnosis [30]. In summary, diagnosis of ONFH requires a multidisciplinary approach to enhance currently limited accuracy and reliability. Prompt diagnosis of ONFH may lead to morbidity and costs avoidance, and due to an increased risk of developing the disease contralaterally, an accurate postoperative pathologic diagnosis may be essential [18].

Study limitations: Albeit the aim of the present work was to underline the key role of multidisciplinary diagnoses of patients with ONFH suspicion, some limitations should be considered before drawing conclusions. The complexity of the process of samples collection, preservation, and preparation allowed us to analyze only twenty-four suitable specimens, and thus, the limitations of a small sample size and that of it being a monocentric study should be considered.

## 5. Conclusions

To enhance current diagnostic precision of ONFH we propose a closer collaboration between clinicians and a greater participation of pathologists. Macroscopic evaluation of the femoral head in the operating room and a more extended use of MRI are also suggested.

Regarding anatomopathological analyses of ONFH, our findings support the quantification of diffuse empty osteocytic lacunae as a valid diagnostic criterion of trabecular necrosis. According to our results, this parameter is not directly related to age or imaging stage significantly. Further investigation of intra- or inter-observer variability in the pathological diagnosis of ONFH is needed.

## Figures and Tables

**Figure 1 diagnostics-12-01731-f001:**
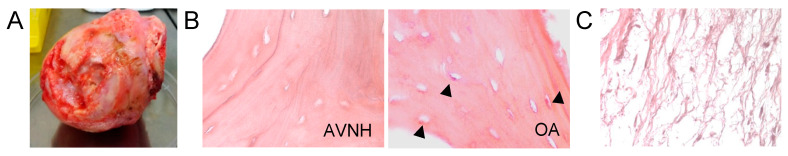
(**A**) Gross specimen of advanced femoral head osteonecrosis with associated osteoarthritis, showing complete loss of articular surface and collapse of the central region of the head. (**B**) Hematoxylin and eosin-stained sections showing trabeculae; (**A**) partially necrotic trabecula: bone showing empty lacunae in ONFH samples. (20×). (**B**) Osteoarthritis (control group) trabecula showing lacunae containing viable osteocytes, necrotic osteocytes, and empty lacunae (20×). (**C**) Hematoxylin and eosin-stained section showing dense medullary fibrosis in ONFH bone samples (20×).

**Table 1 diagnostics-12-01731-t001:** From the initial sixty femoral heads collected, only twenty-four met our selection criteria, twelve with a previous diagnosis of ONFH and twelve controls with an OA diagnosis. The table illustrates the diagnosis of every sample according to different specialists: orthopedic surgeons, radiologists, and pathologists.

Specimen	Surgeon Pre-Surgery	Surgeon Post-Surgery	Radiologist	Pathologist	Disagreement
1 (CASE)	AVNH	AVNH	AVNH	AVNH	NO
2 (CASE)	AVNH	AVNH	AVNH	AVNH	NO
3 (CASE)	AVNH	AVNH	OA	AVNH	Radiologist vs. others
4 (CASE)	AVNH	AVNH	AVNH	AVNH	NO
5 (CASE)	AVNH	AVNH	AVNH	AVNH	NO
6 (CASE)	AVNH	AVNH	AVNH	AVNH	NO
7 (CASE)	AVNH	AVNH	AVNH	AVNH	NO
8 (CASE)	OA	OA	OA	AVNH	Pathologist vs. others
9 (CASE)	OA	OA	AVNH	AVNH	Surgeon vs. others
10 (CASE)	AVNH	AVNH	AVNH	AVNH	NO
11 (CASE)	AVNH	AVNH	AVNH	AVNH	NO
12 (CASE)	OA	OA	OA	AVNH	Pathologist vs. others
13 (CONTROL)	OA	AVNH	OA	OA	Surgeon vs. others
14 (CONTROL)	OA	OA	OA	OA	NO
15 (CONTROL)	OA	OA	OA	OA	NO
16 (CONTROL)	OA	OA	OA	OA	NO
17 (CONTROL)	OA	OA	OA	OA	NO
18 (CONTROL)	OA	OA	OA	AVNH	Pathologist vs. others
19 (CONTROL)	OA	OA	OA	OA	NO
20 (CONTROL)	OA	OA	OA	OA	NO
21 (CONTROL)	OA	OA	OA	OA	NO
22 (CONTROL)	OA	OA	OA	AVNH	Pathologist vs. others
23 (CONTROL)	OA	OA	OA	OA	NO
24 (CONTROL)	OA	OA	OA	OA	NO

**Table 2 diagnostics-12-01731-t002:** Histological analysis data from femoral head specimens.

	ONFH (Mean ± SE)	Control (Mean ± SE)	*p* Value Intergroup Differences
Age (years)	65.88 ± 12.6	63.84 ± 10.9	0.262991
Empty osteocytic lacunae (%)	68.16 ± 10.57	40.91 ± 15.03	0.000019
Fatty infiltration (1–4) *	3.5 ± 0.9	2.83 ± 0.83	0.08914
Medullary fibrosis (1–4) *	2.75 ± 1.21	1.66 ± 0.98	0.03486
Heterotopic ossification (%) ⴕ	25 ± 0.45	25 ± 0.45	1
Age (years)	65.88 ± 12.6	63.84 ± 10.9	-

*: “1-absence”, “2-presence”, “3-moderate”, and “4-intense”. ⴕ: significative presence yes/no.

**Table 3 diagnostics-12-01731-t003:** Data comparing Ficat and Arlet imaging stage and the average number of empty osteocytic lacunae in our ONFH specimens.

Ficat and Arlet Stage	Number of Samples Analyzed	Empty Osteocytic Lacunae (%)
I	2	63
II	1	80
III	3	73
IV	6	65

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
