# Peer review of "Osteonecrosis of the Femoral Head: A Multidisciplinary Approach in Diagnostic Accuracy"

_diagnostics, 2022, doi:10.3390/diagnostics12071731_

Round 1
Reviewer 1 Report
Dear Authors,
besides the intrinsic limitations of monocentric design, in my opinion, the topic is interesting considering the need to increase diagnostic accuracy and reliability in this common condition.
However, I have concerns about the methodological implant of the study and some critical issues should be addressed.
Major revisions
INTRODUCTION. This section should be improved, highlighting the need for a precise diagnosis and staging of AVNH in order to perform the optimal therapeutic approach that might include multimodal conservative treatment or a patient-tailored plan from hip surgery to rehabilitation.
According to this, you should cite the following references:
- Luan S et al. Comparisons of Ultrasound-Guided Platelet-Rich Plasma Intra-Articular Injection and Extracorporeal Shock Wave Therapy in Treating ARCO I–III Symptomatic Non-Traumatic Femoral Head Necrosis: A Randomized Controlled Clinical Trial. Journal of Pain Research 2022;Volume 15:341–54.
- de Sire A et al. Optimization of transdisciplinary management of elderly with femur proximal extremity fracture: A patient-tailored plan from orthopaedics to rehabilitation. World Journal of Orthopedics 2021;12:456–66.
- Sconza C et al. Multimodal conservative treatment of migrating bone marrow edema associated with early osteonecrosis of the hip. SAGE Open Medical Case Reports 2022;10:2050313X2110676.
METHODS. Please, you should clearly present the study’s design in this section.
METHODS. Please, all data and numbers should be presented in the “Results” Section rather than “Methods” Section.
METHODS. Please, you should better define inclusion and exclusion criteria.
METHODS. Please, clarify how the sample size was calculated.
RESULTS. Patients assessed for eligibility and patients excluded should be clarified in the results section, characterizing at least the main cause of exclusions.
DISCUSSION. A specific “Study Limitations” subsection should be provided at the end of Discussion Section, especially discussing the limitations of a small sample size and monocentric design.
Minor revisions
ABSTRACT. Page 1, line 13. Please, correct “is both” with “are both”.
ABSTRACT. Page 1, line 15. Please, correct “exent” with “exempt”.
WHOLE TEXT. Please, use English punctuation avoiding comma for decimal numbers.
INTRODUCTION. Many sentences are not followed by the appropriate reference; please, provide the right source. E.g.: page 1, line 41; page 2, line 53.
RESULTS. Page 3, line 108. Please, correct “procedures[10,11]” with “procedures [10,11]”.
TABLE 1. I suggest formatting the table to separate the AVNH group from the control group.
TABLE 2. Please, add a column with the corresponding P values for intergroup differences.
RESULTS. TABLE 3. Page 5, line 154. This sentence is not clear; please, improve it. Moreover, add the P values for the differences you mention in Table 3.
DISCUSSION. Page 6, line 186. Please, correct “collection.[7]” with “collection [7].”.
DISCUSSION. Please, correct the several spacing mistakes between words and citations’ brackets.
Author Response
Answers to reviewers’ comments.
#Reviewer 1 Comments to the author
Dear Authors,
besides the intrinsic limitations of monocentric design, in my opinion, the topic is interesting considering the need to increase diagnostic accuracy and reliability in this common condition.
However, I have concerns about the methodological implant of the study and some critical issues should be addressed.
We are pleased that the reviewer thought that we are addressing an interesting topic and agrees with us on the clinical need to increase diagnostic accuracy in osteonecrosis of the femoral head. We have addressed all the reviewers’ concerns and introduced the indicated changes in the manuscript, which are now shown in red.
Major revisions
INTRODUCTION. This section should be improved, highlighting the need for a precise diagnosis and staging of AVNH in order to perform the optimal therapeutic approach that might include multimodal conservative treatment or a patient-tailored plan from hip surgery to rehabilitation.
According to this, you should cite the following references:
- Luan S et al. Comparisons of Ultrasound-Guided Platelet-Rich Plasma Intra-Articular Injection and Extracorporeal Shock Wave Therapy in Treating ARCO I–III Symptomatic Non-Traumatic Femoral Head Necrosis: A Randomized Controlled Clinical Trial. Journal of Pain Research 2022;Volume 15:341–54.
- de Sire A et al. Optimization of transdisciplinary management of elderly with femur proximal extremity fracture: A patient-tailored plan from orthopaedics to rehabilitation. World Journal of Orthopedics 2021;12:456–66.
- Sconza C et al. Multimodal conservative treatment of migrating bone marrow edema associated with early osteonecrosis of the hip. SAGE Open Medical Case Reports 2022;10:2050313X2110676.
We agree with the reviewer that perhaps our introduction did not appropriately reflect the need for a precise diagnosis of this invalidating pathology. Following the reviewers’ directions, we have now added a couple of paragraphs to underscore the necessity of having an early and accurate diagnosis of the ONFH, and include the three references indicated by the reviewer in the introduction.
METHODS. Please, you should clearly present the study’s design in this section.
We have now introduced a “Study design” section with all the details concerning this study.
METHODS. Please, all data and numbers should be presented in the “Results” Section rather than “Methods” Section.
We thank the reviewer for pointing this out. All the data and numbers have been moved and are now displayed at the beginning of the Results section.
METHODS. Please, you should better define inclusion and exclusion criteria.
Inclusion and exclusion criteria are now clarified in the Methods section. Besides, all the relevant references regarding these criteria have also been added.
METHODS. Please, clarify how the sample size was calculated.
Although osteonecrosis of the femoral head is not a rare disease, it has a much lower incidence than other musculoskeletal diseases such as osteoporosis or osteoarthritis. Taking this into account, is easy to understand why the studies regarding ONFH do normally include a relatively small sample size compared to that of more prevalent musculoskeletal diseases. We intended for this study to have a sample size similar to other studies in this line in the field of osteonecrosis. We indicate here those studies as well as the Femoral heads with a diagnosis of ONFH used in each of them.
Mukisi-Mukaza, M.; Gomez-Brouchet, A.; Donkerwolcke, M.; Hinsenkamp, M.; Burny, F. Histopathology of aseptic necrosis of the femoral head in sickle cell disease. Int Orthop 2011, 35, 1145-1150, doi:10.1007/s00264-010-1121-6. (11 femoral heads)
Lang P, Jergesen HE, Moseley ME, Block JE, Chafetz NI, Genant HK. Avascular necrosis of the femoral head: high-field-strength MR imaging with histologic correlation. Radiology. 1988 Nov;169(2):517-24. (9 femoral heads)
Simmons DJ, Daum WJ, Totty W, Murphy WA. Correlation of MRI images with histology in avascular necrosis in the hip. A preliminary study. J Arthroplasty. 1989;4(1):7-14.
(5 femoral heads)
Kim YH, Kim JS. Histologic analysis of acetabular and proximal femoral bone in patients with osteonecrosis of the femoral head. J Bone Joint Surg Am. 2004 Nov;86(11):2471-4.
(25 femoral heads)
Humphreys S, Spencer JD, Tighe JR, Cumming RR. The femoral head in osteonecrosis. A quantitative study of osteocyte population. J Bone Joint Surg Br. 1989 Mar;71(2):205-8.
(15 femoral heads)
Plenk H Jr, Gstettner M, Grossschmidt K, Breitenseher M, Urban M, Hofmann S. Magnetic resonance imaging and histology of repair in femoral head osteonecrosis. Clin Orthop Relat Res. 2001 May;(386):42-53. (14 femoral heads)
Yeh LR, Chen CK, Huang YL, Pan HB, Yang CF. Diagnostic performance of MR imaging in the assessment of subchondral fractures in avascular necrosis of the femoral head. Skeletal Radiol. 2009 Jun;38(6):559-64 (25 femoral heads)
RESULTS. Patients assessed for eligibility and patients excluded should be clarified in the results section, characterizing at least the main cause of exclusions.
This information has now been clarified in the results section.
DISCUSSION. A specific “Study Limitations” subsection should be provided at the end of Discussion Section, especially discussing the limitations of a small sample size and monocentric design.
A study limitation subsection has now been introduced at the end of the Discussion section.
Minor revisions
We would like to thank the reviewer for his/her thorough revision of the manuscript and for pointing out these errors. We have now carefully revised the manuscript for spelling and grammatical errors, adjust all the spacing and formatted the tables as suggested, adding information where required. All changes are indicated in red in the uploaded manuscript. We believe this has really improved the manuscript.
ABSTRACT. Page 1, line 13. Please, correct “is both” with “are both”.
ABSTRACT. Page 1, line 15. Please, correct “exent” with “exempt”.
WHOLE TEXT. Please, use English punctuation avoiding comma for decimal numbers.
INTRODUCTION. Many sentences are not followed by the appropriate reference; please, provide the right source. E.g.: page 1, line 41; page 2, line 53.
RESULTS. Page 3, line 108. Please, correct “procedures[10,11]” with “procedures [10,11]”.
TABLE 1. I suggest formatting the table to separate the AVNH group from the control group.
TABLE 2. Please, add a column with the corresponding P values for intergroup differences.
RESULTS. TABLE 3. Page 5, line 154. This sentence is not clear; please, improve it. Moreover, add the P values for the differences you mention in Table 3.
DISCUSSION. Page 6, line 186. Please, correct “collection.[7]” with “collection [7].”.
DISCUSSION. Please, correct the several spacing mistakes between words and citations’

Reviewer 2 Report
Methodological Biases exist
Tables that present the outcomes are missing
Limitations exist
(The Authors must see my remarks)

Author Response
Answers to reviewers’ comments.
#Reviewer 2.
We thank the reviewer for the revision of this manuscript. We believe that have addressed all the reviewers’ concerns and introduced the indicated changes in the manuscript, which are now shown in red.
We must clarify that we have encountered some technical problems with the commented pdf containing the revision that was made available to us. It has been difficult for us to determine the precise part of the text that was linked to the comment since some bubble icons appeared misplaced and not above the text in the version we downloaded. We hope we have been able to pinpoint the commented section for each of the bubble icons but would like to apologise in advance if this technical problem causes any confusion.
We have clarified the type of article as requested by the reviewer and have changed the title using Osteonecrosis of the Femoral Head instead Avascular Necrosis of the Hip, to make clear that we are indeed referring to osteonecrosis, as pointed out by the reviewer.
We agree with the reviewer that some parts of the manuscript lacked appropriated references. The relevant references have now been added to the manuscript, including those referring to the inclusion and exclusion criteria, as requested by the reviewer.
The information about the study (study design) has now been completed and added to the methods section. Regarding the study sample, it must be considered that ONFH is not a common condition. In countries with similar demographics and health determinants to Spain, like Germany or Japan, the incidence of AVNH is around 0.01% (Hofmann S, Kramer J, Plenk H. 2005 Feb;34(2):171-83;). And excluding corticosteroids, bisphosphonates or heavy alcohol intake, as happens when you look for idiopathic cases of the disease, makes numbers even more scarce. We have based the sample size of our study on the range of samples used in similar publications. Here we indicate those studies as well as the femoral heads with a diagnosis of ONFH included in each of them.
Mukisi-Mukaza, M.; Gomez-Brouchet, A.; Donkerwolcke, M.; Hinsenkamp, M.; Burny, F. Histopathology of aseptic necrosis of the femoral head in sickle cell disease. Int Orthop 2011, 35, 1145-1150, doi:10.1007/s00264-010-1121-6. (11 femoral heads)
Lang P, Jergesen HE, Moseley ME, Block JE, Chafetz NI, Genant HK. Avascular necrosis of the femoral head: high-field-strength MR imaging with histologic correlation. Radiology. 1988 Nov;169(2):517-24. (9 femoral heads)
Simmons DJ, Daum WJ, Totty W, Murphy WA. Correlation of MRI images with histology in avascular necrosis in the hip. A preliminary study. J Arthroplasty. 1989;4(1):7-14. (5 femoral heads)
Kim YH, Kim JS. Histologic analysis of acetabular and proximal femoral bone in patients with osteonecrosis of the femoral head. J Bone Joint Surg Am. 2004 Nov;86(11):2471-4. (25 femoral heads)
Humphreys S, Spencer JD, Tighe JR, Cumming RR. The femoral head in osteonecrosis. A quantitative study of osteocyte population. J Bone Joint Surg Br. 1989 Mar;71(2):205-8. (15 femoral heads)
Plenk H Jr, Gstettner M, Grossschmidt K, Breitenseher M, Urban M, Hofmann S. Magnetic resonance imaging and histology of repair in femoral head osteonecrosis. Clin Orthop Relat Res. 2001 May;(386):42-53. (14 femoral heads)
Yeh LR, Chen CK, Huang YL, Pan HB, Yang CF. Diagnostic performance of MR imaging in the assessment of subchondral fractures in avascular necrosis of the femoral head. Skeletal Radiol. 2009 Jun;38(6):559-64 (25 femoral heads)
We agree with the reviewer that Spearmans’ can be an unreliable model in some circumstances. Spearman's Rho is a non-parametric test used to measure the strength of association between two variables, where the value r = 1 means a perfect positive correlation and the value r = -1 means a perfect negative correlation. In our experience, when comparing two histologic parameters we could not find a significant difference, so I am afraid, in this case its transcendence is negligible. To illustrate its use of these methods in our field we enclose a reference we certainly considered helpful in the development of our paper: Zhang, YZ., Cao, XY., Li, XC. et al. Accuracy of MRI diagnosis of early osteonecrosis of the femoral head: a meta-analysis and systematic review. J Orthop Surg Res 13, 167 (2018).
We have added a paragraph pointing out the limitations of the study as requested by the reviewer. This section can be found at the end of the discussion section.
The conclusion section has been reduced and only the main outcomes are stated, as indicated.
Other spelling/grammar mistakes that have been kindly pointed out by the reviewer have also been corrected.
Round 2
Reviewer 1 Report
Dear Authors,
I would like to thank you for your careful revision and for addressing my concerns. I think the manuscript has been further improved during the revision process. I suggest just a minor revision before publication.
Minor revision
INTRODUCTION: Please highlight the role of transdisciplinary management following hip surgery, in accordance with the most recent evidence.
You might write: "Presumably, the optimal therapeutic approach might include a transdisciplinary treatment regime or a patient-tailored plan from hip surgery to rehabilitation"
The following reference might be cited:
- de Sire A, Invernizzi M, Baricich A, Lippi L, Ammendolia A, Grassi FA, Leigheb M. Optimization of transdisciplinary management of elderly with femur proximal extremity fracture: A patient-tailored plan from orthopaedics to rehabilitation. World J Orthop. 2021 Jul 18;12(7):456-466. doi: 10.5312/wjo.v12.i7.456. PMID: 34354934; PMCID: PMC8316838.
Nice work, good luck
Author Response
We sincerely thank the reviewer for his/her thorough revision of the manuscript and for the update in the references.
We have now modified the manuscript according to the reviewers' instructions and hope that the manuscript now meets his/her requirements.
Reviewer 2 Report
Few corrections are required
(The Authors must see my remarks)

Author Response
Answers to reviewers’ comments.
#Reviewer 2 Comments to the author
We thank the reviewer for his/her thorough revision and were happy to see that only minor issues were left unresolved in our previous answer. We apologise for not having meet the reviewers’ requirements in the first round of revision and have done our best to meet those requirements in this second round.
1.-Regarding the study sample, we have now introduced a sentence in the methods section (2.1. Study design) indicating that “the study sample was determined following the guidelines of other studies in this line, made in the field of osteonecrosis” and added up new seven references to support the validity of our sample size”.
2.- In relation to the statistical methods used, in particular the Spearmans’ coefficient, while we agree with this reviewer that, under some circumstances this model is unreliable, we believe that for this particular study, its use is not fully unjustified. Spearman´s rank correlation does not require continuous-level data (interval or ratio), because it uses ranks instead of assumptions about the distributions of the two variables. This allows us to analyse the association between variables of ordinal measurement levels. Moreover, the Spearman correlation does not assume that the variables are normally distributed. A Spearman correlation analysis can therefore be used in many cases in which the assumptions of the Pearson correlation (continuous-level variables, linearity, heteroscedasticity, and normality) are not met. These characteristics explain its wide use in biological, medical and social sciences. To illustrate extended utilisation, we enclose a brief list of recent articles from our own field that have employed it successfully:
- Wei Q, He W, Zhang Q, Chen Z, Zheng Y, Lin T. [Clinical significance of different imaging manifestations of osteonecrosis of femoral head in the peri-collapse stage]. Zhongguo Xiu Fu Chong Jian Wai Ke Za Zhi. 2021 Sep 15;35(9):1105-1110. Chinese. doi: 10.7507/1002-1892.202103221. PMID: 34523274; PMCID: PMC8444142.
- Watanabe N, Murakami S, Uchida S, Tateishi S, Ohara H, Yamamoto Y, Kojima T; JAHORN (Japan Arthroscopy of the Hip Outcomes Research Network). Validity of the Japanese Orthopaedic Association Hip Disease Evaluation Questionnaire (JHEQ) for Japanese patients with labral tear. J Hip Preserv Surg. 2020 Oct 29;7(3):466-473. doi: 10.1093/jhps/hnaa038. PMID: 33948202; PMCID: PMC8081416.
- Chu K, Cheng G, Yu GZ, Ning B, Jia TH. Inconsistency of Bone Mineral Density Between Femoral Head and Proximal Femur After Femoral Neck Fracture Surgery Indicates Great Possibility of Femoral Head Necrosis. Orthopedics. 2021 Mar-Apr;44(2):e223-e228. doi: 10.3928/01477447-20201216-06. Epub 2020 Dec 30. Erratum in: Orthopedics. 2021 Nov-Dec;44(6):332. PMID: 33373461.
- Yi Z, Bo Z, Bin S, Jing Y, Zongke Z, Fuxing P. Clinical Results and Metal Ion Levels After Ceramic-on-Metal Total Hip Arthroplasty: A Mean 50-Month Prospective Single-Center Study. J Arthroplasty. 2016 Feb;31(2):438-41. doi: 10.1016/j.arth.2015.09.034. Epub 2015 Sep 28. PMID: 26515043.
- Wei B, Wei W. Identification of aberrantly expressed of serum microRNAs in patients with hormone-induced non-traumatic osteonecrosis of the femoral head. Biomed Pharmacother. 2015 Oct;75:191-5. doi: 10.1016/j.biopha.2015.07.016. Epub 2015 Aug 19. PMID: 26298803; PMCID: PMC7127261.
- Kuribayashi M, Takahashi KA, Fujioka M, Ueshima K, Inoue S, Kubo T. Reliability and validity of the Japanese Orthopaedic Association hip score. J Orthop Sci. 2010 Jul;15(4):452-8. doi: 10.1007/s00776-010-1490-0. Epub 2010 Aug 19. PMID: 20721711.
- Chen XC, Liu M, Pan YQ, Yu CH, Lu HJ, Du JC, Chen XQ. [Clinical significance of venous return disturbance in patients with osteonecrosis of femoral head]. Zhejiang Da Xue Xue Bao Yi Xue Ban. 2009 Jan;38(1):95-9. Chinese. doi: 10.3785/j.issn.1008-9292.2009.01.014. PMID: 19253435.
- Stöve J, Riederle F, Kessler S, Puhl W, Günther KP. Reproduzierbarkeit radiologischer Klassifikationskriterien der Femurkopfnekrose [Reproducibility of radiological classification criteria of femur head necrosis]. Z Orthop Ihre Grenzgeb. 2001 Mar-Apr;139(2):163-7. German. doi: 10.1055/s-2001-15050. PMID: 11386108.
- Huang G, Zhao G, Xia J, Wei Y, Chen F, Chen J, Shi J. FGF2 and FAM201A affect the development of osteonecrosis of the femoral head after femoral neck fracture. Gene. 2018 Apr 30;652:39-47. doi: 10.1016/j.gene.2018.01.090. Epub 2018 Jan 31. PMID: 29382571.
3.- Finally, regarding the reviewers’ question of “how do we know that?” referred to the sentence “Other non-specific degenerative signs examined, such as fatty infiltration and heterotopic ossification showed no statistically significant difference between the two groups”: The reviewer is absolutely right as we have not shown the results of these analysis that were not included in the previous round of revision by mistake. We deeply apologise for this error. We have now added a supplementary table (supplementary table 1) gathering all the information related to the results of these histological analyses with the relevant information added in the table legend. The reference to this table has been added to the main text (line 219) We hope this helps to clarify this point.